Corrected: Author correction

# Ultra-thin high-efficiency mid-infrared transmissive Huygens meta-optics

Li Zhang[1,2], Jun Ding [3,4], Hanyu Zheng[1,2], Sensong An[4], Hongtao Lin [2], Bowen Zheng[4], Qingyang Du[2], Gufan Yin[2], Jerome Michon[2], Yifei Zhang[2], Zhuoran Fang[2], Mikhail Y. Shalaginov[2], Longjiang Deng[1], Tian Gu[2], Hualiang Zhang[4] & Juejun Hu[2]

The mid-infrared (mid-IR) is a strategically important band for numerous applications ranging from night vision to biochemical sensing. Here we theoretically analyzed and experimentally realized a Huygens metasurface platform capable of fulfilling a diverse cross-section of optical functions in the mid-IR. The meta-optical elements were constructed using high-index chalcogenide films deposited on fluoride substrates: the choices of wide-band transparent materials allow the design to be scaled across a broad infrared spectrum. Capitalizing on a two-component Huygens' meta-atom design, the meta-optical devices feature an ultra-thin profile ($\lambda_0/8$ in thickness) and measured optical efficiencies up to 75% in transmissive mode for linearly polarized light, representing major improvements over state-of-the-art. We have also demonstrated mid-IR transmissive meta-lenses with diffraction-limited focusing and imaging performance. The projected size, weight and power advantages, coupled with the manufacturing scalability leveraging standard microfabrication technologies, make the Huygens meta-optical devices promising for next-generation mid-IR system applications.

[1] State Key Laboratory of Electronic Thin Films and Integrated Devices, National Engineering Research Center of Electromagnetic Radiation Control Materials, University of Electronic Science and Technology of China, Chengdu, Sichuan 611731, China. [2] Department of Materials Science & Engineering, Massachusetts Institute of Technology, Cambridge, MA 02139, USA. [3] School of Information and Science Technology, East China Normal University, Shanghai 200062, China. [4] Department of Electrical & Computer Engineering, University of Massachusetts Lowell, Lowell, MA 01854, USA. These authors contributed equally: Li Zhang, Jun Ding, Hanyu Zheng. Correspondence and requests for materials should be addressed to T.G. (email: gutian@mit.edu) or to H.Z. (email: hualiang_zhang@uml.edu) or to J.H. (email: hujuejun@mit.edu)

The mid-infrared (mid-IR) spectral region (spanning 2.5–10 μm in wavelength) contains the characteristic vibrational absorption bands of most molecules as well as two atmospheric transmission windows, and is therefore of critical importance to many biomedical, military, and industrial applications such as spectroscopic sensing, thermal imaging, free-space communications, and infrared countermeasures. However, devices operating in the mid-IR band often present a technical challenge for optical engineers. Since most traditional optical materials including silicate glasses and optical polymers become opaque at wavelength beyond 3 μm, mid-IR optical components are either made of specialty materials such as chalcogenides or halides whose processing technologies are less mature, or require complicated fabrication methods such as diamond turning (e.g., in the cases of silicon or germanium optics). Consequently, unlike visible or near-infrared optical parts which are commonplace and economically available off the shelf, mid-IR optics are plagued by much higher costs and often inferior performance compared to their visible or near-infrared counterparts.

Optical metasurfaces, artificial materials with wavelength-scale thicknesses and on-demand electromagnetic responses[1–3], provide a promising solution for cost-effective, high-performance infrared optics. These thin-sheet structures can be readily fabricated using standard microfabrication technologies, thereby potentially enabling large-area, low-cost manufacturing. Their singular electromagnetic properties, custom-tailorable through meta-atom engineering, allow optical designers to manipulate unconventional optical behavior (e.g., abnormal refraction/reflection, dispersion compensation, etc.) inaccessible to natural materials at the macroscale. In addition, their planar nature is conducive to "flat" optics and systems with drastically reduced size, weight, and power (SWaP) relative to their traditional bulk equivalents.

So far, a number of metasurface-based functional optical elements have been realized in the mid-IR range, including lenses[4], perfect absorbers[5–8], polarization controllers[9,10], modulators[11,12], vortex beam generators[13], thermal emitters[14], nonlinear converters[15], and infrared sensors[16]. Unfortunately, the devices are based on metallic nanostructures whose large optical losses restrict their operation to the reflective mode. The limitation is eliminated in metasurfaces built entirely out of dielectric materials[17–19], and such transmissive dielectric meta-optics offer several well-established advantages in optical system design including increased alignment tolerance and simplified on-axis configuration. Recently, chiral metasurfaces made of $Si/SiO_2$[20] and meta-lenses fabricated using Si-on-sapphire[21] have been demonstrated. The former device attained a polarization conversion efficiency up to 50%. The latter Si-on-sapphire meta-lens exhibited transmission efficiency as high as 79%, although the optical efficiency was not reported and optical focusing was not demonstrated either. Furthermore, operation bands of both material systems are bound to be below 5 μm wavelength due to onset of phonon absorption in $SiO_2$ or sapphire.

In this article, we report the design and experimental demonstration of high-efficiency mid-IR transmissive optics based on dielectric Huygens metasurface (HMS). The novelty of our work is twofold. First of all, we choose the chalcogenide alloy PbTe to construct the meta-atoms. With its exceptionally high refractive index exceeding 5, PbTe is ideally suited for creating dielectric meta-atoms supporting high-quality Mie resonances[22]. The nanocrystalline nature of PbTe also facilitates monolithic integration on the low-index $CaF_2$ substrate ($n = 1.4$). The large index contrast between the $PbTe–CaF_2$ couple contributes to the ultra-thin profile of the meta-atoms. The material pair also exhibits low optical attenuation from 4 to 9 μm wavelengths, compatible with transmissive metasurface designs across most of the mid-IR band. The material choice further facilitates scalable manufacturing of meta-optics: we have already validated large-area (on full 6" wafers), high-throughput (growth rate ~100 nm/min) PbTe film deposition via simple single-source thermal evaporation and wafer-scale lithographic patterning of the film[23–26], and optical quality $CaF_2$ substrates are now commercially available with diameters up to 4". In addition to the material innovation, our work also marks the first experimental demonstration of HMS in the mid-IR, and claims significant performance improvement over previously demonstrated HMS devices at optical frequencies leveraging an advanced two-component meta-atom design. Unlike dielectric metasurfaces based on waveguiding effects which mandate high aspect ratio nanostructures to cover full $2\pi$ phase[27,28], the concept of Huygens metasurfaces, originally derived from the field equivalence principle and elaborated in Supplementary Note 1, enables exquisite control of electromagnetic wave propagation in a low-profile surface layer with deep sub-wavelength thickness[29]. A remarkable feature of HMS is that near-unity optical efficiency is possible in such a metasurface comprising meta-atoms possessing both electric dipole (ED) and magnetic dipole (MD) resonances[30–32]. Nevertheless, dielectric Huygens optical metasurfaces experimentally demonstrated to date, which are made up of a single type of circular or rectangular shaped meta-atoms with varying sizes, suffer from much lower efficiencies around or below 50%[33–37]. Here we show that such single-component meta-atom constructions incur an inherent trade-off between phase coverage and transmission efficiency, and we overcome this limitation by employing a two-component meta-atom design consisting of both rectangular and H-shaped meta-atom structures. This unique approach significantly boosts the optical transmittance of HMS to above 80% with overall optical efficiencies up to 75%, while maintaining a thin profile with a thickness of $\lambda_0/8$. This unique combination of judicious material choice and innovative HMS design allows us to demonstrate high-performance transmissive meta-optics operating near the mid-IR wavelength of 5.2 μm. In the following, we first discuss the HMS design rationale followed by elaboration of the material characterization and device fabrication protocols. Design and optical characterization results of three meta-optical devices, a diffractive beam deflector, a cylindrical lens, and an aspheric lens are then presented.

## Results

**Huygens metasurface design**. Figure 1a sketches the structure of a rectangular meta-atom in the form of a PbTe block sitting on a $CaF_2$ substrate. The size of a unit cell, $P$, is 2.5 μm along both axes, less than $\lambda_0/2$ to eliminate undesired diffraction orders. Total thickness of the PbTe block is fixed at 650 nm or 1/8 of the free-space wavelength ($\lambda_0 = 5.2$ μm), the smallest among dielectric metasurfaces reported to date.

To ascertain that the rectangular meta-atom indeed supports both electric and magnetic dipole resonances, we simulated the optical transmittance spectra (Supplementary Fig. 2b, Supplementary Note 2) and field profiles at ED and MD resonances (Supplementary Figs. 2c-j). According to the Kerker condition[38], spectrally overlapping ED and MD resonances allow a maximum of $2\pi$ phase shift with near-unity transmittance. To fulfill the condition, the ED and MD resonances of the rectangular meta-atom are independently tuned through adjusting the PbTe block dimensions $L_x$ and $L_y$ (Supplementary Fig. 3): the degrees of freedom enables full $2\pi$ phase coverage by detuning the ED and MD resonances slightly off the operation wavelength. Figure 1b, c charts the numerically simulated transmission phase and amplitude of the rectangular meta-atom as functions of $L_x$ and $L_y$ at $\lambda_0 = 5.2$ μm. Optimal rectangular meta-atom performance,

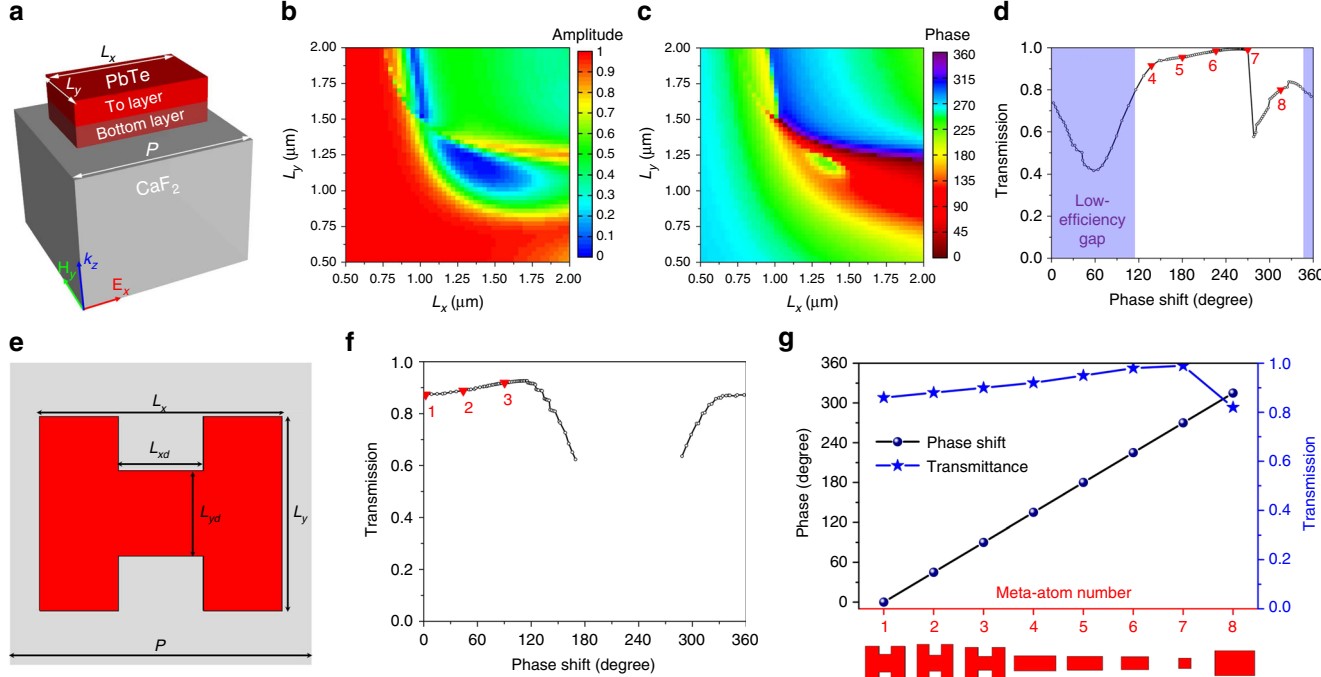

**Fig. 1** Simulation results of meta-atoms. **a** Schematic tilted view of a rectangular meta-atom structure; optical transmission **b** amplitude and **c** phase of the rectangular meta-atom as functions of the meta-atom dimensions; **d** optimized optical transmission of rectangular meta-atoms with different phase delay values: the shaded region corresponds to the "low-efficiency gap" where rectangular meta-atoms fail to provide satisfactory performance; **e** top-view schematic of an H-shaped meta-atom; **f** optimized optical transmission of H-shaped meta-atoms with different phase delay values: the design offers superior efficiency to bridge the "low-efficiency gap"; **g** phase shift and transmittance of the eight meta-atom elements used to construct the meta-optical devices: the corresponding meta-atom designs are also marked in (**d**) and (**f**) with red triangles

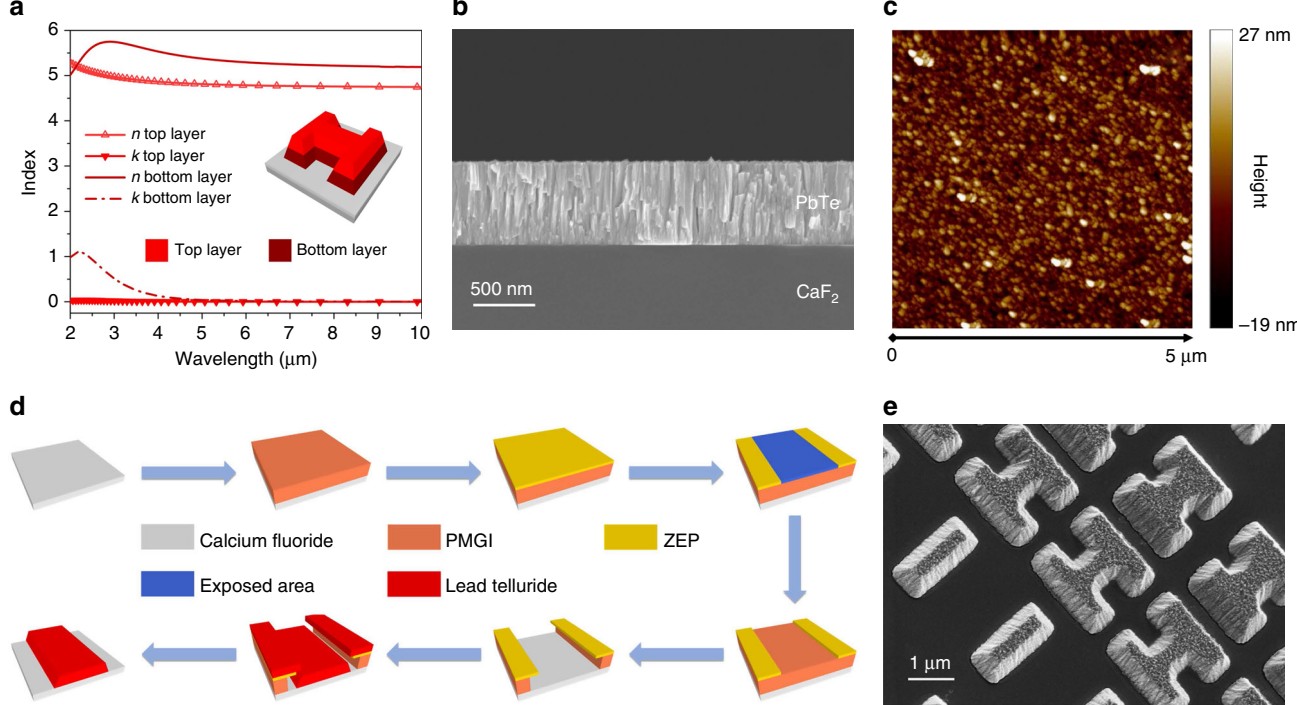

**Fig. 2** Material characterization and fabrication process. **a** Refractive index $n$ and extinction coefficient $k$ of the PbTe material measured using ellipsometry and fitted to a two-layer model; inset shows a schematic depiction of an H-shaped meta-atom; **b** cross-sectional SEM image of the PbTe film; **c** surface morphology of PbTe film measured using AFM; **d** schematic fabrication process flow of the meta-optical devices; **e** tilted-view SEM image of fabricated metasurface structure

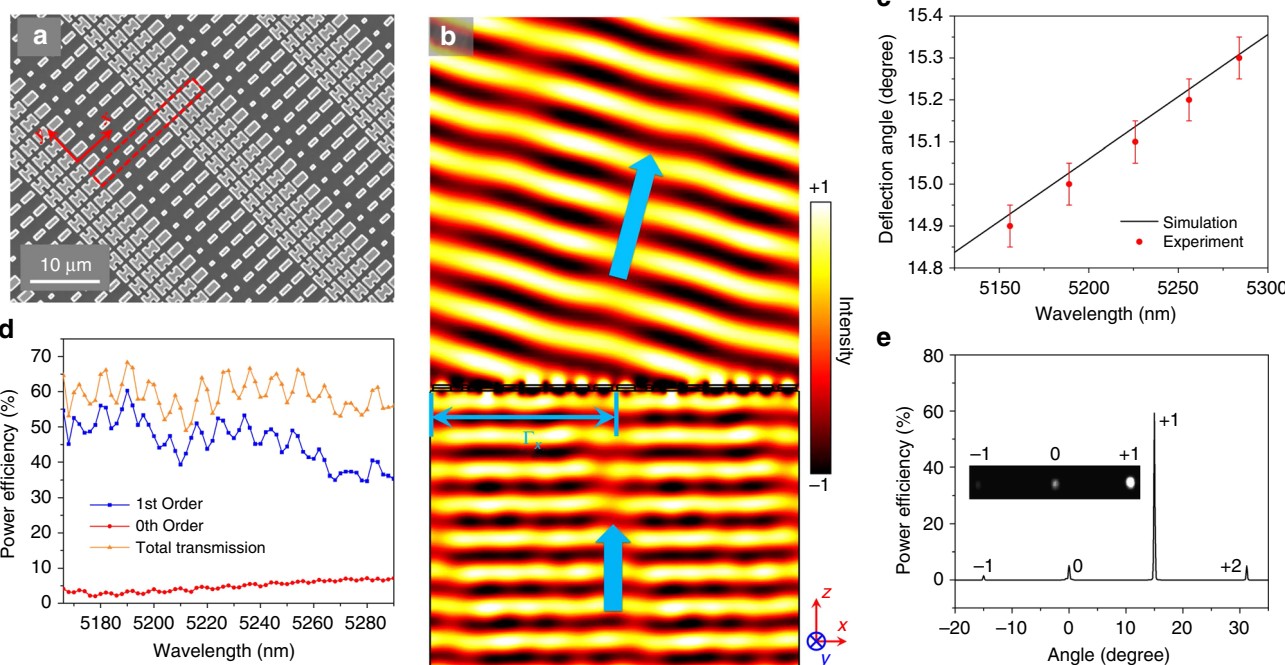

**Fig. 3** Characterization of diffractive beam deflector. **a** Top-view SEM image of the HMS beam deflector: the red box marks the unit cell; **b** simulated electric field profile when a plane wave at 5.2 μm wavelength is incident upon the metasurface from the substrate side, clearly showing the beam deflection effect; **c** simulated and experimentally evaluated wavelength dependence of the beam deflection angle, error bar originated from accuracy (±0.1) of measurement setup; **d** experimentally measured spectra for total transmission (sum of all transmissive diffraction orders), 1st diffraction order (the blazed order) and 0th order (specular transmission) of the deflector; **e** angle-dependent output intensity from the device measured at 5.19 μm wavelength. Inset shows an infrared image of the diffracted beams at the same wavelength

derived from these figures by selecting $L_x$ and $L_y$ to yield the highest optical transmission at each phase shift value between 0 to $2\pi$, is plotted in Fig. 1d. From the figure, we see that while the rectangular meta-atom design offers $2\pi$ phase coverage, ~120° phase range (shaded area in Fig. 1d) comes with low optical transmission (<80%). The observation suggests that at least two unit cells will endure poor optical efficiency if eight discretized phases are adopted for the HMS. Similar low-efficiency gaps can also be observed in HMS designs relying on a single type of meta-atom geometry, be it circular, elliptical, or square[33,34,36].

To circumvent this limitation, we devised a new class of meta-atoms with an H-shaped geometry (Fig. 1e). Campione et al.[39] suggested that the ED and MD of a dielectric resonator can be tailored by introducing air gaps: the H-shaped meta-atom can be deemed as a pair of dielectric resonators separated by an air gap and connected by a dielectric bar. We show that the H-shaped meta-atoms exhibit both ED and MD resonances, and their resonant behavior is readily tuned by varying the dielectric bar dimensions (Supplementary Note 3). Figure 1f plots the simulated transmission amplitude of the H-shaped meta-atoms as a function of the corresponding phase delay with exemplary results illustrated in Supplementary Figs. 4c and 4d, indicating that transmission exceeding 85% can be attained within the entire low-efficiency gap of the rectangular meta-atoms. Our HMS unit cells, illustrated in Fig. 1g, combines the rectangular and H-shaped meta-atoms to achieve superior optical efficiency across the full $2\pi$ phase range. This unique two-component HMS design underlies the unprecedented high performance of our meta-optical devices.

**Material characterization and device fabrication.** PbTe films with a thickness of 650 nm were thermally evaporated onto double-side polished CaF₂ substrates. Figure 2a plots the refractive index $n$ and extinction coefficient $k$ of the PbTe material

measured using variable angle spectroscopic ellipsometry (J.A. Woollam Co.). We found that a phenomenological two-layer model best describes the optical properties of the film, which properly accounts for the slight composition and microstructure variation throughout the film thickness owing to noncongruent vaporization and the columnar growth mechanism[40]. The model also yields excellent agreement between our design and experimental measurements on the meta-optical devices. Figure 2b shows a cross-sectional scanning electron microscopy (SEM) image of the PbTe film, revealing a dense, columnar nanocrystalline microstructure free of voids. The film's fine grain structure produces a smooth surface finish with a root-mean-square (RMS) surface roughness of 6 nm, evidenced by the atomic force microscopy (AFM) image in Fig. 2c. This low surface roughness contributes to minimizing optical scattering loss despite the high index contrast.

Figure 2d schematically depicts the fabrication flow of the meta-optical devices. Details of the process are furnished in the Methods section. In contrast to metasurfaces relying on waveguiding effects which entails meta-atoms with a large aspect ratio and specialized fabrication protocols[41], our rugged, low-profile (with a maximum aspect ratio of merely 1.25) HMS structures can be readily fabricated using a simple double-layer electron beam resist lift-off process. We note that the lift-off process results in PbTe structures with a sidewall angle of 68°, and this non-vertical sidewall profile was taken into consideration in our meta-atom design. We measured RMS sidewall roughness of 12 nm and a small roughness correlation length of 11 nm on the fabricated meta-atoms, which is primarily attributed to the nanoscale columnar grain structure as shown in Fig. 2e. Such roughness incurs negligible scattering loss in the mid-IR meta-optical devices.

**Diffractive beam deflector.** Figure 3a displays a top-view SEM image of the fabricated meta-optical beam deflector. The supercell

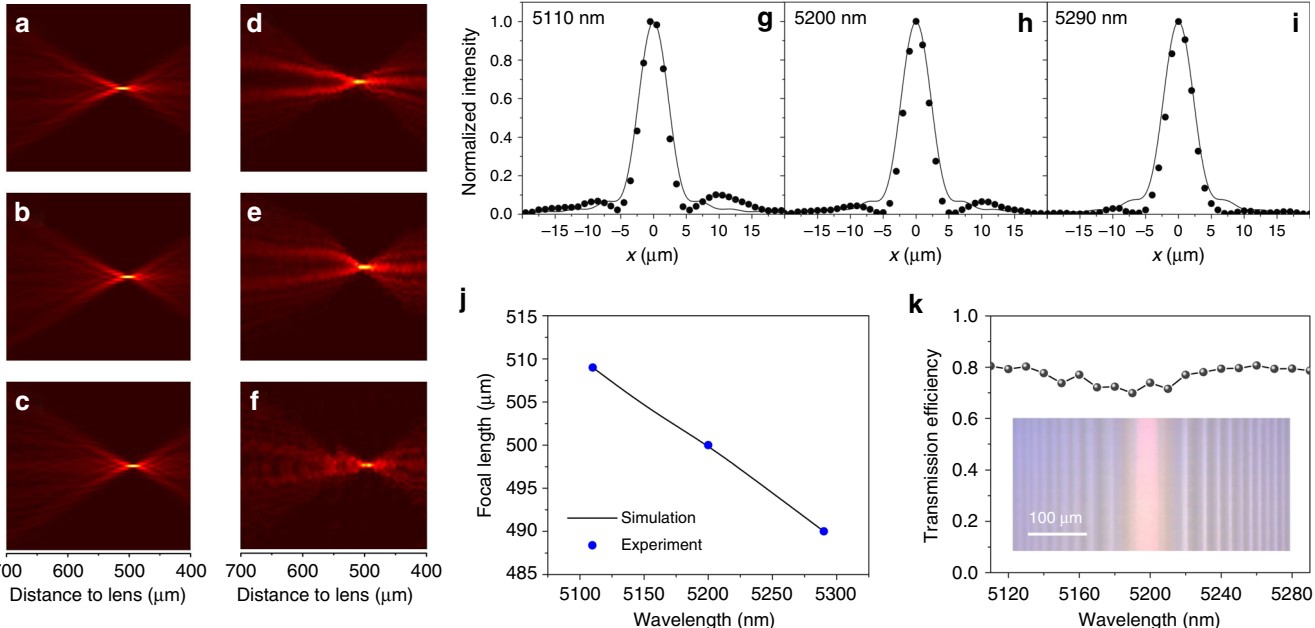

**Fig. 4** Characterization of cylindrical flat lens. **a–c** Simulated and **d–f** measured intensity distributions near the focal spot of the cylindrical lens at the wavelength of **a**, **d** 5110 nm, **b**, **e** 5200 nm, and **c**, **f** 5290 nm; **g–i** intensity profiles at the focal plane at **g** 5110 nm, **h** 5200 nm, and **i** 5290 nm wavelength: the solid lines represent simulated responses from an ideal lens whereas the dots are experimental data; **j** wavelength-dependent focal length of the cylindrical lens; **k** measured transmission efficiency of the cylindrical lens as a function of wavelength; inset shows a top-view optical micrograph of the fabricated meta-lens

(labeled with a red box) consists of eight meta-atoms based on the two-component design illustrated in Fig. 1g. The supercell is tiled along both $x$ and $y$ directions with periods of $\Gamma_x = 20\,\mu m$ and $\Gamma_y = 2.5\,\mu m$, respectively. Along the $x$-axis, the structure acts as a diffractive grating, and the meta-atoms generate a step-wise phase profile resembling that of a traditional blazed grating to selectively enhance optical scattering into the first diffraction order while suppressing all others. Along the $y$-direction, the period is smaller than the free-space wavelength and hence no diffraction (other than the zeroth-order specular transmission) takes place.

Figure 3b portrays the simulated electric field distribution when a plane wave polarized along the $x$-direction at 5.2 μm wavelength is incident on the deflector from the substrate ($CaF_2$) side. According to the theoretical model, most optical power (82%) is concentrated into the first transmissive diffraction order at a Bragg deflection angle of 15.1°. Figure 3c, d plots the measured deflection angle and diffraction efficiency of the deflector over the spectral band of 5.16–5.29 μm. The Fabry–Perot fringes on the measured spectra, which exhibit a free spectral range of 9.3 nm, result from reflections at the 1 mm-thick $CaF_2$ substrate surfaces.

The two key performance metrics of a beam deflector are absolute diffraction efficiency, which is defined as power of the deflected beam in the target (blazed) diffraction order normalized to total incident power (not corrected for Fresnel reflection losses from the substrate surfaces), and extinction ratio (ER), the optical power ratio between the first diffraction order and specular transmission (the zeroth order). We attained a peak absolute diffraction efficiency of 60% and an ER of 12 dB at 5.19 μm wavelength (Fig. 3e). We note that the diffraction efficiency is considerably lower than the model predicted value of 82%, likely due to phase error resulting from imperfect fabrication. Still, the device claims significant performance improvement over previously reported HMS beam deflectors at optical frequencies (diffraction efficiency 20%, ER 3 dB[35], and diffraction efficiency 36%, ER not reported[33]) and is on par with the best mid-IR

transmissive gratings demonstrated to date (diffraction efficiency 63%, ER 11 dB)[42].

**Cylindrical flat lens**. In the HMS cylindrical lens, the meta-atoms are arranged such that a hyperbolic phase profile is introduced along the $x$-axis, whereas the structure is tiled along the $y$-direction with a sub-wavelength period to suppress non-specular diffraction orders. The lens is designed with a focal length of 0.5 mm and a numerical aperture (NA) of 0.71: both design parameters were validated experimentally (Supplementary Note 5). Figure 4a–c presents the modeled optical intensity profile when the cylindrical meta-lens is illuminated by an $x$-polarized plane wave at the wavelengths of 5.11, 5.2, and 5.29 μm. In comparison, Fig. 4d–f plots the corresponding experimental measurement results at these wavelengths, which agree well with the simulations. To gauge focusing quality of the lens, line scans were performed to map the optical intensity distributions at the focal plane. Figure 4g–i compares the data with computed intensity profiles for a one-dimensional focused beam at the diffraction limit: the excellent agreement indicates that our HMS lens exhibit diffraction-limited focusing performance. The longitudinal chromatic aberration (focal length change per unit wavelength) extracted from the line scan profiles amounts to −0.11 μm/nm, which agrees perfectly with modeling outcome (Fig. 4j).

Measured optical transmittance through the lens is approximately 80% (normalized to total incident light power without Fresnel loss correction) across the spectral range of 5.11–5.29 μm (Fig. 4k). Unfortunately, the lens focusing efficiency (defined as optical power at the focal spot/line divided by the total incident power) could not be assessed experimentally, since mid-IR cylindrical lenses with an NA equal to or larger than 0.71 (the NA of our HMS lens) that allows the photodetector to fully capture the focused light beam were not commercially available. Nevertheless, we anticipate that the focusing efficiency is likely close to 80% since minimal optical scattering was observed, as suggested

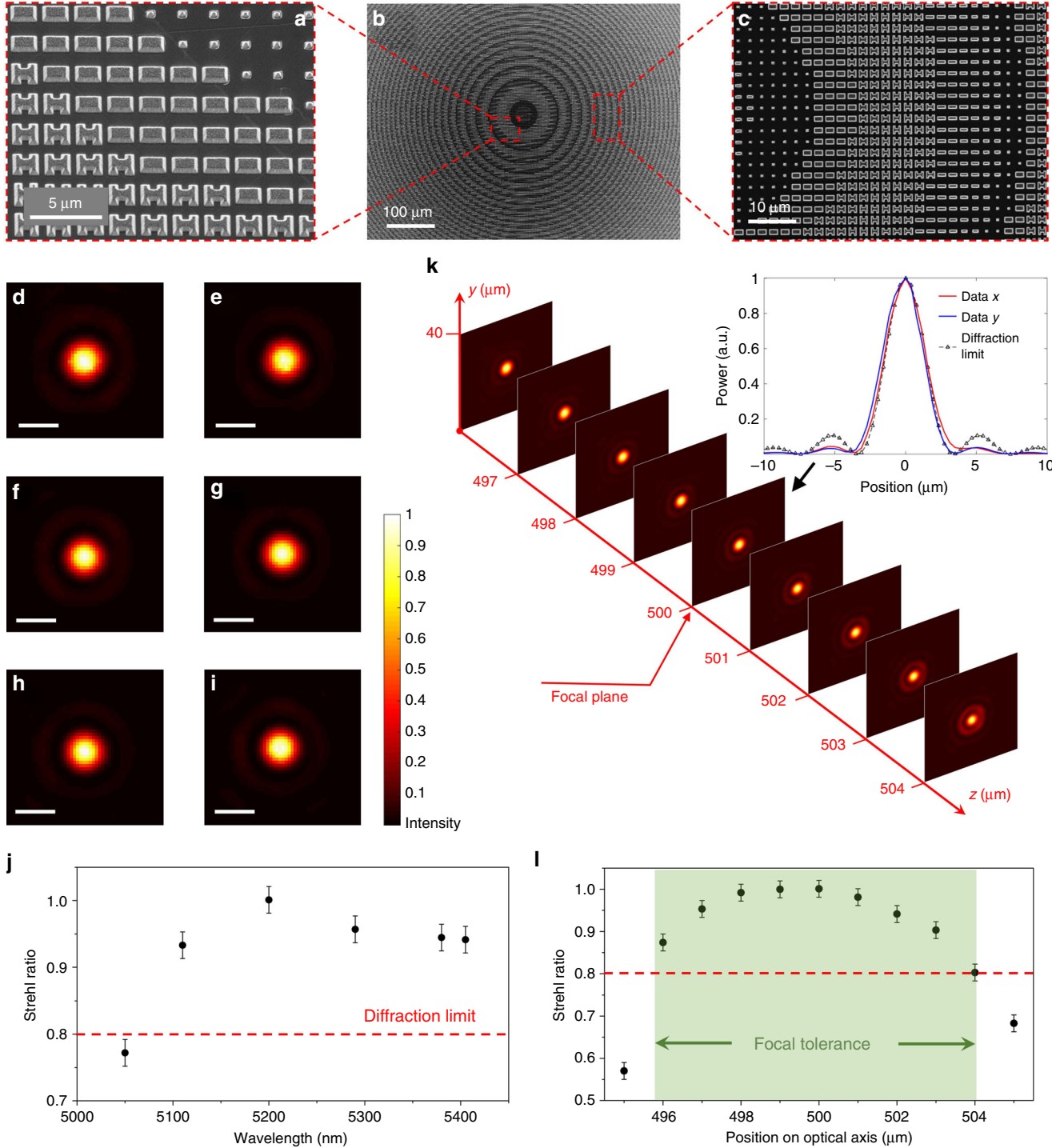

**Fig. 5** Characterization of aspherical meta-lens. **a–c** Top-view SEM micrographs of the fabricated aspheric meta-lens; **d–i** measured aspheric lens focal spot profiles at the wavelengths of **d** 5050 nm, **e** 5110 nm, **f** 5200 nm, **g** 5290 nm, **h** 5380 nm, and **i** 5405 nm; the scale bar represents 5 μm; **j** measured Strehl ratios as a function of wavelength; **k** focal spot profile evolution along the optical axis at 5200 nm wavelength; inset: measured intensity distributions of the focal spot at 5200 nm wavelength along *x*- and *y*-directions on the focal plane in comparison with diffraction-limited focal spot profile. The measurement data are normalized such that the total power on the focal plane (rather than peak intensity) equals that of the diffraction-limited focal spot; **l** Strehl ratios measured on the optical axis: the shaded region corresponds to focal tolerance of the meta-lens. Error bars for panels **j** and **l** are explained in Supplementary Note 7

by data in Fig. 4d–f. Similarly high focusing efficiency was verified on two-dimensional aspheric lenses presented in the following section, which adopts the identical Huygens meta-atom design.

**Aspheric meta-lens**. Aspheric lenses are essential elements for aberration-free imaging systems. Metasurfaces, with their facile

ability to generate almost arbitrary optical phase profiles, provide a versatile alternative to conventional geometric shaping in aspheric lens design. Metasurface-based aspheric lenses with focusing performances at the diffraction limit have been demonstrated in the telecom band[43] and in the mid-IR with a reflectarray configuration[4]. Here we experimentally realized, for

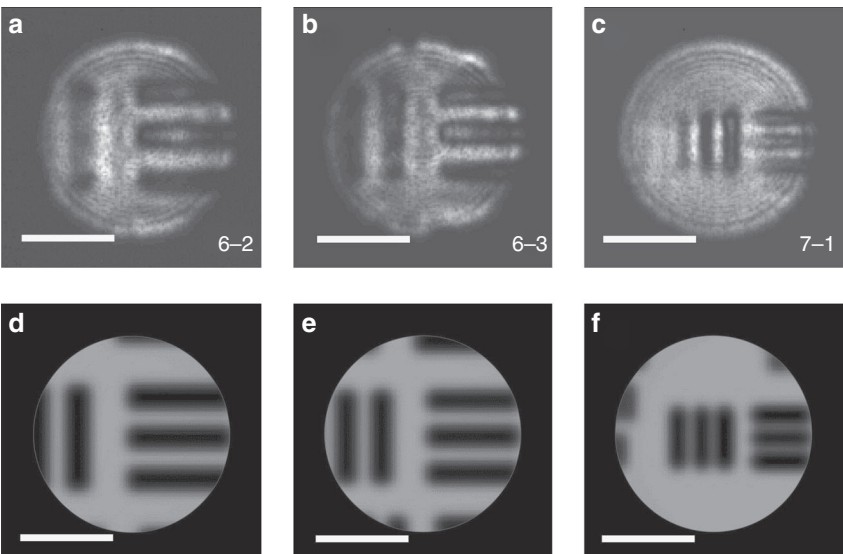

**Fig. 6** Image testing results. **a–c** Optical images of USAF-1951 resolution targets collected using the aspheric meta-lens as a microscope objective: the numbers mark the group numbers of the bar target elements; **d–f** simulated images of the same groups of bar targets acquired by a hypothetical aberration-free imaging system otherwise identical to the experimental setup. The scale bars correspond to 30 μm

the first time, diffraction-limited focusing capability in a mid-IR transmissive metasurface lens.

Figure 5a–c presents top-view images of the HMS aspheric lens structure. The lens assumes a hyperbolic phase profile at 5.2 μm wavelength, which is discretized on an $x$–$y$ grid and implemented using the two-component meta-atoms. According to the design, the lens has a 1 mm by 1 mm square aperture and a focal length of 0.5 mm at 5.2 μm wavelength. Focusing characteristics of the lens were examined throughout the entire tuning range of our laser. The lens focusing efficiency was consistently measured at approximately 75%. The transverse focal spot profiles at multiple wavelengths are shown in Fig. 5d–i. Modulation transfer functions (MTFs) of the lens, computed from Fourier transform of the focal spot profiles[44], are presented in Supplementary Fig. 12 alongside the MTFs of an ideal diffraction-limited lens of the same aperture size. Figure 5j plots the Strehl ratio of the meta-lens (Supplementary Note 7): the ratio is above 0.8 from 5.11 to 5.405 μm wavelengths. The close resemblance of the measured MTFs with those of the ideal lens, combined with the near-unity Strehl ratio, prove diffraction-limited focusing performance of the HMS lens across the 5.11–5.405 μm band.

Focal tolerance of the meta-lens, defined as the on-axis range within which the peak intensity is above 80% of that at the focal plane[45], was inferred from the focal spot profiles along the optical axis (Fig. 5k). Strehl ratios calculated from the results indicate that the lens exhibits a focal tolerance of 8 μm at 5.2 μm wavelength (Fig. 5l).

We further characterized imaging properties of the meta-lens. Figure 6a–c shows images of USAF-1951 resolution targets collected by an infrared microscope using the meta-lens as the objective at 5.2 μm wavelength. Figure 6d–f presents simulated images of the same groups of bar targets generated by a hypothetical aberration-free imaging system otherwise identical to the experimental setup. In our experiment, the microscope can readily resolve bar targets with a gap width of 3.9 μm (USAF-1951 Group 7, Element 1). This is consistent with a theoretical resolution limit of 3.4 μm specified by the Rayleigh criterion[45]. The results confirm that the HMS lens can indeed offer near-diffraction-limit, sub-wavelength imaging capability.

In summary, we have designed and experimentally implemented a transmissive Huygens meta-optics platform operating in the

mid-IR. The devices we realized leveraging this platform uniquely combine a thin, deep sub-wavelength profile and superior optical performances, which benefit from the ultra-high index and broadband transparency of chalcogenide materials, as well as the optically efficient two-component Huygens meta-atom design. The suite of meta-optical devices we demonstrated include diffractive beam deflectors with an absolute diffraction efficiency of 60% and an extinction ratio of 12 dB, high-NA cylindrical lenses with a transmission efficiency of 80%, and aspheric lenses with a focusing efficiency of 75%, all of which represent substantial improvement over state-of-the-art mid-IR meta-optics. We also achieved, for the first time, diffraction-limited focusing in the mid-IR using transmissive meta-lenses. Sub-wavelength imaging was also demonstrated using our devices (during revision cycle of the article, we learnt about the following related work which demonstrated wavelength-scale imaging using mid-IR metasurfaces[46]). Compared to traditional mid-IR optical elements which are bulky and often cost prohibitive, the remarkable optical performance, SWaP advantages, and fabrication scalability of the meta-optics platform foresee their widespread adoption in future infrared optical systems.

## Methods

**Device modeling**. The full-wave electromagnetic simulations were performed using the commercial software package CST Microwave Studio. For single-cell (meta-atom) design, the unit cell boundary condition was employed for the calculations of transmission and phase shift in an array structure. When modeling the meta-optical devices, open boundary is set in the $z$-axis and an $x$-polarized plane wave was illuminated from the substrate side. For the deflector, periodical boundary conditions (PBC) are set in both $x$- and $y$-directions; for the cylindrical lens, PBC is set along the $y$-axis and open boundary (add space) is used in the $x$-axis; and for the aspheric lens, open boundary is implemented in both $x$- and $y$-axes.

The focusing and imaging behavior of ideal diffraction-limited systems (as a comparison to the measured performance in our meta-optical elements) was modeled following the Kirchhoff diffraction integral, a physically rigorous form of the Huygens–Fresnel principle[45]. The model starts with computing the Huygens point spread function of the optical system. It converts each ray from the source into a wavefront with amplitude and phase, which then propagates to the image plane where its complex amplitude is derived. The diffraction of the wavefront through space is given by the interference or coherent sum of the wavefronts from the Huygens sources. The intensity at each point on the image plane is the square of the resulting complex amplitude sum. The technique was applied to simulate the diffraction-limited focal spot profiles in Fig. 4g–i, Fig. 5k inset, and the Strehl ratio calculations (Fig. 5j, l), as well as the aberration-free images in Fig. 6d–f. In

computing Fig. 6d–f, the compound lens in Supplementary Fig. 9b was assumed to be ideal without aberration.

**Device fabrication.** Device fabrication was performed at the Harvard Center for Nanoscale Systems. The starting substrates for the meta-optical devices are double-side polished calcium fluoride (CaF$_2$) from MTI Corporation. Prior to lithographic patterning, the substrates were treated with oxygen plasma to improve adhesion with resist layers and PbTe thin films. A polymethylglutarimide (PMGI-SF9, MicroChem Corp.) resist layer with a thickness of 800 nm was first spin coated on the substrate, followed by coating of a ZEP-520A electron beam resist film (Zeon Chemicals L.P.) with a thickness of 400 nm. A water-soluble conductive polymer layer (ESpacer 300Z, Showa Denko America, Inc.) was subsequently coated on the resist film to prevent charging during electron beam writing, and the polymer layer was removed after lithography by rinsing in deionized water[47]. The ZEP resist was exposed on an Elionix ELS-F125 electron beam lithography system. The double-layer resist, ZEP and PMGI, was then sequentially developed by immersion in ZED-N50 (Zeon Chemicals L.P.) and RD6 (Futurrex Inc.) solutions. Development time in the RD6 solution was timed to precisely control undercut of the PMGI layer and facilitate lift-off patterning. The PbTe film was then deposited via thermal evaporation using a custom-designed system (PVD Products, Inc.)[48]. Small chunks of PbTe with a purity of 99.999% (Fisher Scientific) were used as the evaporation source material. The deposition rate was monitored in real time using a quartz crystal microbalance and was stabilized at 17 Å/s. The substrate was not actively cooled although the substrate temperature was maintained below 40 °C throughout the deposition as measured by a thermocouple. After deposition, the devices were soaked in an n-methyl-2-pyrrolidone (NMP) solution heated to 70 °C to lift-off the resist masked portion and complete the device fabrication.

**Device characterization.** The fabricated HMS devices were optically characterized using an external cavity tunable quantum cascade laser (Daylight Solution, Inc.) operating in a mode-hop-free continuous wave mode with x-polarized output. The testing configurations are detailed in Supplementary Note 4 for the beam deflector, Supplementary Note 5 for the cylindrical lens, and Supplementary Note 6 for the aspheric lens. All measurements were performed at room temperature.

**Data availability.** All relevant data in this paper and supplementary materials are available from the corresponding author on reasonable request.

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

## Acknowledgements

J.D. and H. Zhang acknowledge initial funding for design of the meta-atoms provided by the National Science Foundation under award CMMI-1266251. Z.L. and H. Zheng contributed to the Device Fabrication section and were independently funded as visiting scholars by the National Natural Science Foundation of China under award 51772042 and the "111" project (No. B13042) led by Professor Huaiwu Zhang. Later work contained within the Device Modeling and Device Characterization sections and some revisions to the manuscript were funded under Defense Advanced Research Projects Agency Defense Sciences Office (DSO) Program: EXTREME Optics and Imaging (EXTREME) under Agreement No. HR00111720029. The authors also acknowledge fabrication facility support by the Harvard University Center for Nanoscale Systems funded by the National Science Foundation under award 0335765. The views, opinions and/or findings expressed are those of the authors and should not be interpreted as representing the official views or policies of the Department of Defense or the U.S. Government.

## Author contributions

J.D. and H. Zhang conceived the device concepts. H. Zheng and L.Z. fabricated the devices. H. Zheng and T.G. performed device characterizations. J.D., S.A., and B.Z. designed the meta-optical components. T.G., H. Zheng, and J.H. analyzed the experimental data. H.L., Q.D., Y.Z., and Z.F. assisted in device processing. L.Z., G.Y., J.M., and M.S. contributed to optical testing. J.H., T.G., and H. Zhang supervised and coordinated the project. H. Zheng, J.D., T.G., H. Zhang, and J.H. wrote the manuscript. All authors contributed to technical discussions regarding this work.

## Additional information

**Competing interests:** The authors declare no competing interests.

