## [Peer Review File · Nature Communications]

Reviewers' comments:

Reviewer #2 (Remarks to the Author):

This manuscript is a revised version of a previous submission to a different NPG journal. In it, the authors have attempted to address the concerns of the previous two reviewers.

In both the response to the reviewers, as well as the revised manuscript, the authors address many, if not most, of the reviewer concerns.

I believe that this version of the manuscript is suitable for publication essentially as is (perhaps one last check for grammar/typos).

Reviewer #3 (Remarks to the Author):

Here, authors used meta-atoms made of chalcogenide to demonstrate highly efficient metasurfaces in Mid-IR range. Despite the high index of meta-atoms, by properly designing the building blocks' size and geometry they realize high transmission and simultaneously full-phase coverage. I agree with both review #1 and #2 about little new science in this paper: fabrication is very straight forward, materials already are used in previous works and Huygens metasurfaces have been extensively studied before. However, I believe this paper technologically sounds. In particular, achieving efficient metasurface with very simple fabrication, no need for etching, is one of the key factor of this platform which significantly expands its potential applications. I also have several concerns:

1. Similar to previously published Huygens metasurfaces, in simulation building block was an array of meta-atoms, periodic boundary condition, however in deflector case there is no periodicity along the grating period and for spherical lens no periodicity at all. How the initial periodic boundary condition used to calculate the phase is valid and how this will affect the Huygens condition, high transmission, when a meta-atom is used in a device, e.g. lens.

2. Diffraction angle of the demonstrated grating translates to $NA=0.25$ where an efficiency of 66% is achieved. I am very surprised that for the cylindrical lens with a higher $NA=0.71$ than the grating, reported focusing efficiency is 80%. It is well-accepted that higher the NA lower the efficiency. In their reply to reviewer #1, authors try to explain it by saying the grating efficiency considering both orders is higher than the lens which is not a valid argument. The lens efficiency comes from one order not both orders, if we see the lens as a grating with varying period from the center of the edge. This high efficiency needs to be clarified. If cannot be measured, the claim of 80% focusing efficiency for cylindrical lens should be removed. Same question goes to aspherical lens which its focusing efficiency despite having much larger $NA (>0.7)$ is higher than of the grating as well. In addition, grating is a periodic structure and to achieve high efficiency one just needs to suppress zero order but a lens is an aperiodic structure and transmitted light path is not bounded to any specific order which makes lens performance very sensitive to the achieved phase. Therefore, focusing efficiency of a lens can be potentially very different from its transmission efficiency.

[Redacted]

Some minor comments:

a. I agree with first reviewer that the use of CaF₂ is not novel and need to be remove from the claim of novelty.

b. K values, Figure 2a, versus wavelength should be plotted in a separate figure for both substrate and meta-atom material since in current form readers have no idea of the values, most of them look zero.

c. There is not too much of difference between aspect ratio of this work and Si based Mid-IR metasurfaces. I suggest author do not emphasize on the low AR of their design which is misleading.

d. Paper reported a RMS sidewall roughness of 12 nm, however in SEM it looks much larger (Figure 2e), it could be an optical illusion, I suggest that authors provide a zoom-in view SEM of a single meta-atom.

Response to Reviewer Comments NCOMMS-17-24355-T

Reviewer #2 (Remarks to the Author):

This manuscript is a revised version of a previous submission to a different NPG journal. In it, the authors have attempted to address the concerns of the previous two reviewers.

In both the response to the reviewers, as well as the revised manuscript, the authors address many, if not most, of the reviewer concerns.

I believe that this version of the manuscript is suitable for publication essentially as is (perhaps one last check for grammar/typos).

Response:

We thank the reviewer for his/her comments. We have also proofread the manuscript to correct typos following the reviewer's suggestion.

Reviewer #3 (Remarks to the Author):

Here, authors used meta-atoms made of chalcogenide to demonstrate highly efficient metasurfaces in Mid-IR range. Despite the high index of meta-atoms, by properly designing the building blocks' size and geometry they realize high transmission and simultaneously full-phase coverage. I agree with both review #1 and #2 about little new science in this paper: fabrication is very straight forward, materials already are used in previous works and Huygens metasurfaces have been extensively studied before. However, I believe this paper technologically sounds. In particular, achieving efficient metasurface with very simple fabrication, no need for etching, is one of the key factor of this platform which significantly expands its potential applications. I also have several concerns:

1. Similar to previously published Huygens metasurfaces, in simulation building block was an array of meta-atoms, periodic boundary condition, however in deflector case there is no periodicity along the grating period and for spherical lens no periodicity at all. How the initial periodic boundary condition used to calculate the phase is valid and how this will affect the Huygens condition, high transmission, when a meta-atom is used in a device, e.g. lens.

Response:

The comment is related to how to correlate the simulated unit meta-atom performance to that of a functional optical device consisting of arrays of such unit cells. We address this issue by examining the effects of interaction between meta-atoms as well as the re-constructed wave front emitted by the meta-atom array. Optical coupling between the high-index meta-atoms is sufficiently weak such that the phase delay generated by individual meta-atoms is not significantly altered by its neighboring environment.

Fig. R1. Field intensity around the eight Huygens meta-atoms simulated using FDTD

Figure R1 presents the simulated optical field intensity profiles around the meta-atoms when they are excited by light at 5.2 μm wavelength. As can be seen from the figures, the field is primarily confined within the meta-atoms (except for meta-atom #7). The strong confinement accounts for the weak inter-atom coupling in the HMS devices. To further verify that such inter-atom coupling is insignificant in our devices, we modeled the phase profile of light exiting the deflector (solid line in Fig. R2) and compare it with the phase delay of individual meta-atoms extracted from the periodic-boundary simulation (Fig. 1g in the main text). The excellent agreement between the two confirms that the meta-atom phase delay is insensitive to the neighboring atom configurations – even for meta-atom #7.

Fig. R2. Simulated phase profile of light exiting the deflector (red line) and the phase delay imparted by individual meta-atoms according to Fig. 1g.

We further validated the conclusion in a meta-lens structure consisting of 31×31 meta-atoms ($77.5 \mu\text{m} \times 77.5 \mu\text{m}$) with a focal length of $25 \mu\text{m}$ ¹, corresponding to a NA of 0.84. Figure R3 compares the focal spot profiles simulated (a) by assuming that the metasurface functions as a phase mask with no coupling between the meta-atoms, and the spatially-dependent phase mask is defined by the transmittance and phase delay values of individual meta-atoms at each location (see further details in our response below to the second question); and (b) via full-wave simulation using the CST package. The excellent agreement

¹ We choose to model the small lens in light of our limited computational power for full-wave simulations.

between the results again confirms minimal coupling effects between meta-atoms as well as validity of the phase mask approach when applied to our HMS devices.

Fig. R3. (a, b) Optical intensity at the focal plane for the NA = 0.84 meta-lens simulated using: (a) the Kirchhoff integral, which assumes that the metasurface behaves as a phase mask whose phase distribution and transmittance are defined by the individual meta-atom properties (without considering inter-atom coupling); (b) the full-wave CST package, which automatically takes into account inter-atom coupling; (c) comparison of focal spot intensity distribution along the x-axis modeled using the two methods.

2. Diffraction angle of the demonstrated grating translates to $NA=0.25$ where an efficiency of 66% is achieved. I am very surprised that for the cylindrical lens with a higher $NA=0.71$ than the grating, reported focusing efficiency is 80%. It is well-accepted that higher the NA lower the efficiency. In their reply to reviewer #1, authors try to explain it by saying the grating efficiency considering both orders is higher than the lens which is not a valid argument. The lens efficiency comes from one order not both orders, if we see the lens as a grating with varying period from the center of the edge. This high efficiency needs to be clarified. If cannot be measured, the claim of 80% focusing efficiency for cylindrical lens should be removed. Same question goes to aspherical lens which its focusing efficiency despite having much larger NA (>0.7) is higher than that of the grating as well. In addition, grating is a periodic structure and to achieve high efficiency one just needs to suppress zero order but a lens is an aperiodic structure and transmitted light path is not bounded to any specific order which makes lens performance very sensitive to the achieved phase. Therefore, focusing efficiency of a lens can be potentially very different from its transmission efficiency.

Response:

We do thank the reviewer for bringing up this interesting issue.

First of all, we have carefully double checked our experimental measurement protocols and the measured data, and we therefore stand by the results.

Second, we want to clarify that the 80% number quoted is the transmission efficiency for the lens rather than focusing efficiency. This number is measured by directly placing a calibrated large-area detector behind the lens to capture all transmitted light, and we expect little experimental error from this very straightforward test. We therefore will use this what we considered to be highly reliable number to cross validate different metasurface models as detailed below. As we explained in our previous response, the focusing efficiency for the cylindrical lens is unfortunately experimentally difficult to obtain, so we only quote the measured transmission efficiency. The focusing efficiency is likely around 75% based on the asphere lens measurement result.

Third, we also noticed an error in our prior simulation results for the beam deflector due to incorrect time-domain truncation of the impulse-response function. The simulated efficiency for the deflector is now 82% instead of 66% at 5.2 μm , and we have revised the manuscript to correct the mistake.

[Redacted]

[Redacted]

With a validated modeling tool at hand, we can now examine the impact of fabrication error on the measured efficiency values of the HMS deflector and the lens. For that we modeled the optical efficiency (deflection efficiency for deflectors and focusing efficiency for lens) of the

deflector with 15° deflection angle and the lens with $NA = 0.71$ using the previously discussed [Redacted] approach. A set of randomly generated phase errors are added to the 8 constituent meta-atoms to simulate deviations due to fabrication imperfections, which are then used to populate the metasurface devices as we did in our experiments. The results are summarized in Fig. R6, where each dot represents an 8-level HMS device (black for deflector and red for lens) with some meta-atom phase error. The blue line is a polynomial fit of the simulated data. We see that the deflector and the lens exhibit

almost identical sensitivity to phase error despite their different beam deflection angles, consistent with our results above. Following the plot, we see

Fig. R6. Simulated optical efficiencies of the deflector and the lens with varying degrees of meta-atom phase errors

that the average phase error resulting in the 60% measured deflector efficiency is ~ 27 degrees, whereas the average phase error for a 75% efficient metalens is ~ 17 degrees. The result suggests that our deflector indeed suffers from a larger phase error compared to the lens. While we are not entirely certain where the difference comes from, we want to mention that the deflector was the first HMS device we fabricated while the metalens was processed about three months later after several other fabrication iterations. Even though the general fabrication protocols are nominally identical, there were minor improvements as we continually streamlined our process and added to the technical know-how, which could have resulted in the enhanced pattern fidelity. In the revised manuscript, we have acknowledged this issue.

[Redacted]

[Redacted]

243
244

245 Some minor comments:

246

247 a. I agree with first reviewer that the use of CaF₂ is not novel and need to be remove from the
248 claim of novelty.

249

250 Response:

251 We concur with the reviewers and have removed the claims.

252

253 b. K values, Figure 2a, versus wavelength should be plotted in a separate figure for both substrate
254 and meta-atom material since in current form readers have no idea of the values, most of them
255 look zero.

256

257 Response:

258 We want to clarify that Figure 2a does not present data pertaining to the substrate. The CaF₂
259 substrate is a commercial off-the-shelf product and its optical properties are well documented (e.g.
260 <https://refractiveindex.info/?shelf=main&book=CaF2&page=Malitson>). The two curves in Fig. 2a
261 correspond to the n & k values of the two laminates within the PbTe film. It is also important to
262 mention that the actual film is not laminated (as is shown in Fig. 2b). The slight composition and
263 microstructure variation due to the columnar growth mechanism⁷ along the film thickness
264 prompted us to model the film phenomenologically as a two-layer structure. The k value is indeed
265 very low at 5.2 μm (~ 0.01) and accounts for minimal optical absorption given the small thickness
266 of our metasurface. We have revised the text to clarify the confusion.

267

268 c. There is not too much of difference between aspect ratio of this work and Si based Mid-IR
269 metasurfaces. I suggest author do not emphasize on the low AR of their design which is misleading.

270

271 Response:

272 We have toned down our statements about the aspect ratio following the reviewer's suggestion.

273 The Si-based mid-IR metasurface (e.g. in Ref. 21) requires meta-atoms with an aspect ratio of 4.3

274 while in our case it is 1.25. While the over-3-fold difference might not appear to be very significant,
 275 it is worth pointing out that empirically structures with an aspect ratio greater than 2 are not
 276 amenable to lift-off fabrication due to pinching⁸. An alternative solution based on multi-cycle
 277 etching/lift-off had been devised to resolve the challenge⁸, although it would be more
 278 straightforward to stick with simple lift-off. In addition, the low aspect ratio also contributes to
 279 reduced shadowing as we discussed earlier and might play a role in realizing efficient high-NA
 280 meta-lenses.

281
 282 d. Paper reported a RMS sidewall roughness of 12 nm, however in SEM it looks much larger
 283 (Figure 2e), it could be an optical illusion, I suggest that authors provide a zoom-in view SEM of
 284 a single meta-atom.

285
 286 Response:

Fig. R7. Example of meta-atom roughness quantification: (a) a zoom-in top-view image of a meta-atom; (b, c) the image processed by software to (b) locate the meta-atom edges; and (c) generate a binary bitmap; (d) the binary bitmap exported to a spreadsheet to determine the edge profile and roughness statistics.

287
 288 The RMS value was quantified using the protocols below. Figure R7a shows a zoom-in SEM top-
 289 view of a single meta-atom. An image processing software was then used to automatically locate
 290 the meta-atom edges (Fig. R7b) and converted the image to a binary profile (Fig. R7c). Due to the
 291 columnar microstructure of the PbTe film, the roughness is well approximated as line-edge
 292 roughness and therefore the edge roughness is a good representation of the meta-atom surface
 293 roughness. The edge profiles from the binary image (labeled with the red boxes) was subsequently
 294 exported to a spreadsheet, which allowed us to then determine the RMS roughness following the

295 definition. The procedures outlined above were repeated for 10 meta-atoms spread across our
296 sample to obtain an average roughness value, which is what we quote in the manuscript.
297

298 **References**

- 299 1 Lalanne, P. & Chavel, P. Metalenses at visible wavelengths: past, present, perspectives. *Laser*
300 *Photonics Rev* **11**, 1600295 (2017).
- 301 2 Lalanne, P., Astilean, S., Chavel, P., Cambriil, E. & Launois, H. Design and fabrication of blazed
302 binary diffractive elements with sampling periods smaller than the structural cutoff. *JOSA A* **16**,
303 1143-1156 (1999).
- 304 3 Born, M. & Wolf, E. Principles of Optics, 7th (expanded) ed. *Cambridge University Press*,
305 *Cambridge, UK*, 890 (1999).
- 306 4 Yu, N. *et al.* Light propagation with phase discontinuities: generalized laws of reflection and
307 refraction. *Science* **334**, 333-337 (2011).
- 308 5 Epstein, A. & Eleftheriades, G. V. Huygens' metasurfaces via the equivalence principle: design
309 and applications. *J Opt Soc Am B* **33**, A31-A50 (2016).
- 310 6 Pfeiffer, C. & Grbic, A. Metamaterial Huygens' surfaces: tailoring wave fronts with reflectionless
311 sheets. *Phys Rev Lett* **110**, 197401 (2013).
- 312 7 Mazor, A., Srolovitz, D., Hagan, P., Bukiet & BG. Columnar growth in thin films. *Phys Rev Lett*
313 **60**, 424 (1988).
- 314 8 Liu, S. *et al.* Realization of tellurium-based all dielectric optical metamaterials using a multi-cycle
315 deposition-etch process. *Appl Phys Lett* **102**, 161905 (2013).
- 316

REVIEWERS' COMMENTS:

Reviewer #3 (Remarks to the Author):

I very much enjoyed reading the revised version and responses to reviewers and believe the paper is ready for publication.